# 6-Shogaol Inhibits Advanced Glycation End-Products-Induced IL-6 and ICAM-1 Expression by Regulating Oxidative Responses in Human Gingival Fibroblasts

**DOI:** 10.3390/molecules24203705

**Published:** 2019-10-15

**Authors:** Kohei Nonaka, Mika Bando, Eijiro Sakamoto, Yuji Inagaki, Koji Naruishi, Hiromichi Yumoto, Jun-ichi Kido

**Affiliations:** Department of Periodontology and Endodontology, Institute of Biomedical Sciences, Tokushima University Graduate School, Tokushima 770-8504, Japanbanchi@tokushima-u.ac.jp (M.B.); yumoto@tokushima-u.ac.jp (H.Y.)

**Keywords:** advanced glycation end-products, 6-shogaol, ROS, antioxidant enzyme, IL-6, ICAM-1, human gingival fibroblasts

## Abstract

Advanced glycation end-products (AGEs) cause diabetes mellitus (DM) complications and accumulate more highly in periodontal tissues of patients with periodontitis and DM. AGEs aggravate periodontitis with DM by increasing the expression of inflammation-related factors in periodontal tissues. 6-Shogaol, a major compound in ginger, has anti-inflammatory and anti-oxidative activities. However, the influence of shogaol on DM-associated periodontitis is not well known. In this study, the effects of 6-shogaol on AGEs-induced oxidative and anti-oxidative responses, and IL-6 and ICAM-1 expression in human gingival fibroblasts (HGFs) were investigated. When HGFs were cultured with 6-shogaol and AGEs, the activities of reactive oxygen species (ROS) and antioxidant enzymes (heme oxygenase-1 [HO-1] and NAD(P)H quinone dehydrogenase 1 [NQO1]), and IL-6 and ICAM-1 expressions were investigated. RAGE expression and phosphorylation of MAPKs and NF-κB were examined by western blotting. 6-Shogaol significantly inhibited AGEs-induced ROS activity, and increased HO-1 and NQO1 levels compared with the AGEs-treated cells. The AGEs-stimulated expression levels of receptor of AGE (RAGE), IL-6 and ICAM-1 and the phosphorylation of p38, ERK and p65 were attenuated by 6-shogaol. These results suggested that 6-shogaol inhibits AGEs-induced inflammatory responses by regulating oxidative and anti-oxidative activities and may have protective effects on periodontitis with DM.

## 1. Introduction

Diabetes mellitus (DM) is a major risk factor of periodontal diseases [1]. Hyperglycemia occurs in DM and induces high glycation of proteins by a Maillard reaction, and then glycated proteins result in advanced glycation end-products (AGEs) [2]. AGEs accelerate diabetic complications, including diabetic nephropathy, diabetic cataract, diabetic peripheral neuropathy, and cardiovascular disorders [3]. AGEs bind to a receptor for AGE (RAGE) and increase the expression of pro-inflammatory molecules such as interleukin (IL)-1β, IL-6, tumor necrosis factor (TNF)-α, and vascular cell adhesion molecule-1 in some cells [4]. AGEs accumulate excessively in periodontal tissues of DM patients compared with non-DM individuals [5] and were detected in epithelial cells, fibroblasts, endothelial cells, and inflammatory cells in periodontal tissues of DM patients [6]. AGEs levels in serum from diabetic monkey increased depend on an aggravation of periodontal diseases [7]. RAGE was also expressed in gingival tissues of DM patients [8] and RAGE mRNA expression was increased by AGEs in human gingival fibroblasts (HGFs) [9]. AGEs inhibited collagen synthesis [10] and increased the expression of matrix metalloproteinase-1 in HGFs [11]. We showed previously that AGEs increased reactive oxygen species (ROS) production and the expression levels of IL-6 and ICAM-1 via RAGE in HGFs [12].

Oxidative stress is caused by a loss of homeostatic balance between ROS and antioxidant enzymes, and it is known to aggravate periodontal diseases [13,14,15] as well as several inflammatory diseases. Although ROS, including superoxide radical, hydroxyl radical, singlet oxygen, and hydrogen peroxide, are produced continuously as natural by-products of the normal metabolism of oxygen, ROS coupled with an oxidative stress and excessive accumulation of ROS influences the pathogenesis of some diseases, including vascular diseases, diabetes, renal ischemia, atherosclerosis, and periodontal diseases [16,17,18,19].

AGEs induced oxidative stress and increased ROS levels in some cells including human umbilical vein endothelial cells [20], human endothelial cell line (EA.hy926) [21] and thyroid follicular epithelial cells [22]. ROS levels in plasma from patients with periodontitis and type 2 DM were significantly higher than those of healthy individuals, and positively correlated with the value of probing pocket depth in patients with both diseases [23]. These reports suggested that AGEs aggravate inflammation and the destruction of periodontal tissues and bone resorption by inducing oxidative stress and regulating the expression of pro-inflammatory cytokines in DM-associated periodontitis.

ROS levels were increased in neutrophils of periodontal tissues with periodontitis [24], and ROS production in neutrophils was elevated by stimulation with *Porphyromonas gingivalis* (*P.gingivalis*) [25], and ROS induced the degradation of extracellular matrix and stimulate bone resorption by osteoclasts [26,27], and ROS is implicated in inflammation and the destruction of periodontal tissues with periodontitis [13].

In contrast, anti-oxidative responses are induced against oxidative stress with ROS in cells and tissues, protecting tissues against destruction caused by oxidative stresses [14], and antioxidants and ROS are balanced in normal physiological conditions [15]. Heme oxygenase 1 (HO-1) and NAD(P)H quinone dehydrogenase 1 (NQO1) are known as antioxidant enzymes and counteract ROS production. [28,29]. Lipopolysaccharide (LPS) from *P.gingivalis* decreased the expression levels of HO-1 and nuclear transcription factor-erythroid 2-related factor 2 (Nrf2) in a rat periodontitis model [30]. AGEs elevated the levels of HO-1 and NQO1 mRNAs and HO-1 expression in bovine aortic endothelial cells [31]. However, the roles of HO-1 and NQO1 as antioxidants in periodontitis with DM are not well known.

Ginger is the rhizome of the plant *Zingiber officinale* Roscoe and it is widely used as a spice and herbs [32]. The major components of ginger are gingerol and shogaol. Shogaol is a dehydrated form of gingerols and prepared from the dried and thermally treated root, and 6-shogaol is most abundant component in shogaol extract [33]. Shogaols and gingerols have multiple pharmacological efficacies including anti-inflammatory, anti-diabetic, anti-cancer, anti-oxidant, anti-microbial and anti-allergic effects. [34]. 6-Shogaol specifically inhibits the expressions of IL-6, TNF-α and prostaglandin E_2_ by suppressing the LPS-activated Akt/IKK/NF-κB pathway in mouse microglial cells [35]. In addition, 6-shogaol inhibited ROS production in a human polymorphonuclear neutrophils (PMNs) [36] and increased HO-1 expression in human hepatoma cell line (HepG2) [37], and 6-shogaol-rich extract from ginger also up-regulated the expression levels of HO-1 and Nrf2 via the p38 mitogen-activated protein kinase (MAPK) pathway in HepG2 cells [38].

6-Shogaol significantly decreased blood glucose levels in streptozotocin-induced diabetic mice [39], and significantly reduced the levels of diabetic markers such as blood glucose and hemoglobin A1c (HbA1c) and decreased the levels of pro-inflammatory cytokines including TNF-α, IL-6, and monocyte chemoattractant protein (MCP)-1 in blood and the kidney, and further restored Nrf2 expression in the kidney of db/db mice [40]. Although 6-shogaol has anti-diabetic and anti-inflammatory effects, the exact effect of 6-shogaol on periodontitis with DM has not yet been elucidated.

In the present study, we investigated the effects of 6-shogaol on AGEs-induced oxidative and anti-oxidative responses and on AGEs-upregulated IL-6 and ICAM-1 expression in HGFs.

## 2. Results

### 2.1. Effects of 6-shogaol on Cell Viability and Morphology of HGFs

When HGFs were cultured with 6-shogaol (2.5–15 μM) for 48 h, the cell viability of HGFs was not significantly influenced (Figure 1A). Cell culture with 2.5–15 μM 6-shogaol for 48 h did not affect cellular morphology (Figure 1B). Therefore, 2.5–15 μM 6-shogaol was used for the subsequent experiments.

### 2.2. 6-Shogaol Inhibits AGEs-induced ROS Production in HGFs

ROS production in HGFs increased depending on the culture times of 12, 24, and 48 h. AGEs (500 μg/mL) increased ROS production from 12 h of cell culture, and elevated ROS levels by approximately 5-fold at 24 h (Figure 2A,B). When HGFs were cultured with 6-shogaol and AGEs for 12 h, *2.5 μM* 6-shogaol did not significantly inhibited AGEs-induced ROS production, however, 5–15 μM 6-shogaol significantly inhibited this ROS induction (Figure 2A). In contrast, 2.5–15 μM 6-shogaol also significantly suppressed AGEs-induced ROS production when cultured for 24–48 h (Figure 2B,C). After 24 h of culture, 15 μM 6-shogaol decreased AGEs-induced ROS level to approximately 59% (Figure 2B).

### 2.3. 6-Shogaol Increases the Levels of HO-1 and NQO1 in HGFs

The effect of 6-shogaol on anti-oxidative factor levels was investigated in cultures with AGEs (500 μg/mL) for 12 h. Although AGEs had no effect on HO-1 production in HGFs, 5 μM 6-shogaol in the presence of AGEs significantly increased HO-1 levels, and 10 μM 6-shogaol elevated HO-1 levels to approximately 4.2-fold compared with in cells cultured with only AGEs (Figure 3A). Although AGEs (500 μg/mL) significantly inhibited NQO1 activity, 6-shogaol (5 and 10 μM) significantly restored AGEs-downregulated NQO1 activity, and its activity returned to the control level (bovine serum albumin [BSA]) (Figure 3B).

### 2.4. Effects of 6-Shogaol on AGEs-Induced RAGE Expression and Antivations of MAPKs and NF-κB Signaling Pathways in HGFs

AGEs (500 μg/mL) elevated RAGE expression in HGFs, and 6-shogaol canceled AGEs-upregulated RAGE expression in HGFs (Figure 4A). When the effects of 6-shogaol and AGEs on the phosphorylation of MAPKs (p38 and ERK) and NF-κB p65 were investigated, AGEs (500 μg/mL) upregulated the phosphorylation of p38 and ERK MAPKs and NF-κB p65 (Figure 4B–D). This effect of AGEs was similar to our previous study [11]. 6-Shogaol (10 μM) decreased AGEs-stimulated phosphorylation of p38 and ERK MAPKs and NF-κB p65 (Figure 4B–D). When the intensity of each band of three separate samples was analyzed by densitometric measurement, AGEs significantly increased the phosphorylation levels of p38 and ERK MAPKs and NF-κB p65, and 6-shogaol significantly decreased AGEs-upregulated phosphorylation levels.

### 2.5. 6-Shogaol Inhibits AGEs-Induced IL-6 and ICAM-1 Expression in HGFs

AGEs (500 μg/mL) significantly increased IL-6 expression levels by approximately 2.4-fold compared with BSA, and 6-shogaol (5 and 10 μM) significantly inhibited AGE-induced IL-6 levels to almost the control (BSA) level, and the downregulation of IL-6 by 6-shogaol was similar to the inhibitory effect of *N*-acetyl-l-cysteine (NAC, 1 mM), an inhibitor of ROS (Figure 5A). 6-Shogaol (5 μM) significantly decreased ICAM-1 production upregulated by AGEs (500 μg/mL), and 10 μM 6-shogaol suppressed ICAM-1 levels to the control level. This inhibitory effect was similar to that of NAC (Figure 5B).

## 3. Discussion

AGEs, a major factor in DM complications, is thought to influence DM-associated periodontitis by regulating oxidative stress and IL-6, and ICAM-1 expression levels in HGFs [12]. In the present study, 6-shogaol inhibited AGEs-induced inflammation-related responses including ROS, IL-6, and ICAM-1. Ginger components such as 6-shogaol and 6-gingerol have been investigated as therapeutic treatment agents for diabetes-associated complications [40,41,42]. 6-Shogaol and 6-gingerol trapped methylglyoxal (MGO), inhibiting the formation of MGO-induced AGEs [43], and the combination of 6-shogaol and epicatechin more strongly trapped MGO in mouse urine [44], suggesting that 6-shogaol not only suppressed AGEs-induced ROS production and IL-6 expression, but also inhibited a formation of AGEs by trapping MGO.

This component in ginger suppressed ROS production in several cells such as PMNs [36], endothelial cells [45], and epidermal keratinocytes [46], whereas 6-shogaol increased the levels of antioxidants. When human epidermal keratinocytes were cultured with 6-shogaol (10 μM), Nrf2 was positively translocated into the nucleus, NQO1 levels were elevated, and dexamethasone-suppressed HO-1 levels were restored [46]. Another study also showed that 6-shogaol and a 6-shogaol-rich extract from ginger increased the translocation of Nrf2 into the nucleus and enhanced HO-1 levels in a human hepatoma cell line [38]. Considering our study and others’ reports, 6-shogaol appears to inhibit AGEs-induced IL-6 and ICAM-1 expression by regulating oxidative and anti-oxidative responses in HGFs.

The effects of 6-shogaol on oxidative stress and inflammatory responses are similar to those of several plants and their extracts. Green and black tea extracts significantly inhibited AGEs-induced ROS production and IL-6 secretion in 3T3-L1 preadipocytes [47], and Moutan Cortex extract suppressed AGEs-induced IL-6 and monocyte MCP-1 levels in vivo and in vitro [48], and Ginkgo biloba extract suppressed AGEs-induced ROS production in rat kidney fibroblasts [49]. These plant extracts are thought to have anti-diabetic potentials via their anti-oxidative activities [47,48,49]. We speculate that 6-shogaol may also contribute to the treatment for diabetes-associated complications because 6-shogaol has similar effects to other plant extracts.

Although ginger components show several anti-diabetic actions, such as reducing the levels of blood glucose and HbA1c, and decreasing the production of ROS, IL-6, and TNF-α [36,39,40,41], their mechanisms of action are not well known. 6-Shogaol inhibited AGEs-induced RAGE expression in HGFs in the present study, and it inhibited MGO-induced RAGE expression and increased HO-1 expression in human retinal epithelial cells [50], suggesting that 6-shogaol partially functions in a regulator of RAGE expression. We showed previously that AGEs increased the expression levels of IL-6 and ICAM-1 via the RAGE, MAPKs (ERK and p38) and NF-κB pathways in HGFs [12]. When the effects of 6-shogaol on the phosphorylation of p38 and ERK MAPKs and NF-κB p65 were investigated in HGFs, 6-shogaol inhibited the AGEs-induced phosphorylation of these proteins. 6-Shogaol also increased the phosphorylation of p38 and ERK MAPKs, and slightly elevated JNK phosphorylation in human breast cancer cell [51], and specific inhibitors of p38 (SB202190) and PI-3K (LY294002) suppressed 6-shogaol-induced Nrf2 translocation into the nucleus in human colon cancer cells [52]. In the present study, 6-shogaol significantly decreased AGEs-induced ROS production in HGFs, however, its inhibition was not complete. Park et al. [46] also showed that 6-shogaol partially inhibited TNF-α+INF-γ-induced ROS production in human keratinocytes. We did not know the exact reason of partial inhibitory effect of 6-shogaol on ROS production. Since the regulation of ROS production by AGEs and 6-shogaol are associated with multiple factors including MAPK (p38, ERK, JNK), NF-κB and PI3K [12,51,52,53], 6-shogaol may partially inhibit AGEs-induced ROS production. Ginger extract inhibited NF-κB activity and decreased IL-8 and vascular endothelial growth factor secretion in an ovarian cancer cell line [54], and *Zingiber officinale* decreased nuclear NF-κB levels in the livers of high-fat diet-fed rats and IL-1β-induced IL-6 and IL-8 expression levels by downregulation of NF-κB activity in human hepatocytes [55]. Ginger extracts show anti-inflammatory effects through the RAGE, MAPKs, and NF-κB pathways in metabolic diseases and cancers [12,51,52,54,55]. In contrast, oxidative and anti-oxidative pathways are related to the shogaol-induced anti-inflammatory responses, because 6-shogaol and ginger extracts suppressed ROS production and elevated the levels of anti-oxidative factors such as Nrf2, HO-1, and NQO1 levels [36,37,38,45,46]. Kim et al. [37] showed that 6-shogaol exerts antioxidant activity via signaling pathways related to HO-1, Nrf2, and γ-glutamylcysteine synthetase. 6-Shogaol inhibited ROS production and activated HO-1, NQO1, and Nrf2 in vitro and ameliorated experimental allergic dermatitis-like skin lesions in vivo [46], and it suppressed Nrf2 expression in the kidney and decreased the levels of blood glucose and HbA1c in diabetic mice [40]. In HGFs, 6-shogaol inhibited AGEs-induced IL-6 and ICAM-1 expression levels, and NAC, a ROS inhibitor, also showed similar inhibitory effects. 6-Shogaol may produce anti-inflammatory and anti-diabetic actions by modulating oxidative and anti-oxidative responses in periodontal tissues.

Ginger extracts and 6-shogaol exert anti-diabetic effects including a decrease in blood glucose and HbA1c levels, and have preventive potential for DM therapy and amelioration of diabetic complications [39,40,41]. Ginger extracts may have therapeutic effects on several inflammatory diseases: 6-shogaol reduced chronic inflammation in the knees [56] and attenuated ulcerative colitis in a murine model [57], and gingerol prevented both joint inflammation and destruction in experimental rheumatoid arthritis [58]. Furthermore, mouthwash containing ginger extracts showed significant improvement of modified gingival index and gingival bleeding index in patients with gingivitis [59], and 10- and 12-gingerols showed effective anti-bacterial activity for *P.gingivalis* and *Prevotella intermedia* [60].

Taken together, 6-shogaol inhibited AGEs-induced inflammation-related factors by regulating oxidative and anti-oxidative activities in HGFs. Ginger extracts including 6-shogaol may be effective for the prevention and treatment of DM-associated periodontitis.

## 4. Materials and Methods

### 4.1. AGEs and Reagents

AGEs were prepared in accordance with the modified method of Takeuchi et al. [61]. Briefly, 50 mg/mL of BSA (Sigma-Aldrich, St. Louis, MO, USA) and 0.1 mol/L of dl-glyceraldehyde (Sigma-Aldrich) were mixed in sterile phosphate-buffered saline (PBS, 0.2 mol/L, pH 7.4) with penicillin (100 U/mL) and streptomycin (100 μg/mL), and incubated at 37 °C for 7 days. The mixture was dialyzed against PBS (pH 7.4) at three times for 3 days to remove low-molecular-weight reactants and free glyceraldehyde. Non-glycated BSA was prepared as a control solution from the mixture without glyceraldehyde in the same conditions. AGE activity was assayed by determining the fluorescence of AGE and non-glycated BSA solutions at excitation/emission wavelengths of 370/440 nm with reference to the method of Huang et al. [62], and AGE solution showed 45-fold stronger fluorescence intensity than that of control non-glycated BSA solution was used in the present study.

Dulbecco’s modified Eagle’s medium (DMEM) and 6-shogaol were obtained from Wako Pure Chemical Industries (Osaka, Japan), and USDA approved fetal bovine serum (FBS) was from Biosera (Kansas City, MO, USA). Anti-RAGE antibody was obtained from Abcam (#ab54741, Cambridge, UK), and antibodies against p38 (anti-phospho-p38 MAPK antibody: #4631, anti-p38 antibody: #9212), ERK (anti-phospho-p44/42 MAPK antibody: #4376, anti-p44/42 MAPK antibody: #9102), p65 (anti-phospho-NF-κB p65 antibody: #3033, anti-NF-κB p65 antibody: #8242) were from Cell Signaling Technology (Beverly, MA, USA), and anti-β-actin antibody (#A2066) and *N*-acetyl-l-cysteine (NAC) were purchased from Sigma-Aldrich.

### 4.2. Cell Culture

The HGF cell line, CRL-2014^®^, was obtained from ATCC (Manassas, VA, USA). HGFs were seeded at 4,800 cells/cm^2^ and cultured in DMEM supplemented with 10% FBS for 5 days and reached sub-confluency. HGFs were cultured using a 96-well plate (SUMITOMO BAKELITE, Tokyo, Japan) for 5 days and used for cell viability assays. In ROS assays, HGFs were cultured using a 96-well black plate (Nunclon^™^, Thermo Scientific^™^, Waltham, MA, USA) for 5 days, and then treated for ROS assays. In other experiments, the sub-confluent HGFs were pre-treated with 2.5–15 μM 6-shogaol for 1 h and cultured with 500 μg/mL of AGEs or BSA for 0.5–48 h and used for a western blotting analysis and enzyme-linked immunosorbent assays (ELISAs).

### 4.3. Cell viability Assay and Observation of Cellular Morphology

Cell viability was examined using a Cell Counting Kit-8 (CCK-8; Dojindo, Kumamoto, Japan) in accordance with the manufacturer’s instructions. Briefly, HGFs were cultured with 2.5–15 μM 6-shogaol for 48 h after reaching sub-confluency, and then incubated with 10 μL CCK-8 solution at 37 °C for 2 h in a humid atmosphere with 5% CO_2_. The absorbance of each well was measured at 450 nm using a microplate reader (iMark™; Bio-Rad, Hercules, CA, USA). The morphologies of cells treated with 6-shogaol (2.5–15 μM) for 48 h were observed using a phase contrast microscope at 40-fold magnification.

### 4.4. ROS Measurement

ROS was measured using an OxiSelect™ Intracellular ROS Assay kit (Cell Biolabs, San Diego, CA, USA) in accordance with our previously reported method [12]. Briefly, HGFs were cultured in a 96-well black plate for 5 days and incubated with 2′,7′-dichlorofluorescin diacetate (*DCF*, 20 μmol/L) at 37 °C for 1 h. Cells were washed in PBS and stimulated with 2.5–15 μM 6-shogaol in the presence of AGEs (500 μg/mL) or BSA (500 μg/mL) for 12-48 h. The fluorescence *intensity of DCF* in the cultured cells was determined at 480/530 nm using the Varioskan™ Flash Multimode Reader (Nunclon^™,^ Thermo Scientific™).

### 4.5. HO-1 and NQO1 Measurement

Sub-confluent HGFs were pre-treated with 6-shogaol (5 or 10 μM) for 1 h and cultured with AGEs (500 μg/mL) or BSA (500 μg/mL) for 12 h. HO-1 and NQO1 levels were determined using a Human Heme Oxygenase 1 (HO 1) Simple Step ELISA Kit (Abcam) and an NQO1 activity assay Kit (Abcam), respectively, in accordance with the manufacturer’s instructions. Briefly, the lysate from treated cells was collected in cell extraction buffer from the HO-1 kit, incubated on ice for 20 min, and centrifuged at 18,000× *g* for 20 min at 4 °C. HO-1 in the supernatant responded with the antibodies and then with TBS substrate, and the absorbance of the reaction solution was measured at 450 nm. HO-1 levels were normalized to the total cell protein amount, which was measured using Bio-Rad protein assay reagent (Bio-Rad). NQO1 activity in the cell lysate samples was determined by following the reduction of menadione and the simultaneous reduction of water soluble tetrazolium salts (WST1), and expressed as dicoumarol-sensitive activity using the NQO1 kit. Briefly, the treated HGFs were solubilized in the extraction buffer from the kit, incubated on ice for 15 min and centrifuged at 18,000 × *g* for 20 min at 4 °C. NQO1 activity in the supernatant samples was determined and expressed as a percentage of activity in the BSA group.

### 4.6. Western Blotting

Sub-confluent HGFs were pre-treated with 6-shogaol (10 μM) for 1 h and then cultured with 500 μg/mL of AGEs or BSA for 30 min (for MAPKs and NF-κB phosphorylation assays) and 24 h (for RAGE western blotting analysis), and the cell lysates were extracted in a lysis buffer including 10 mmol/l Tris- HCl, pH 7.4, 50 mmol/l NaCl, 5 mmol/l EDTA, 1 mmol/l sodium orthovanadate, 1% NP-40 and protease inhibitor cocktail (Complete™; Roche Diagnosis, Berkeley, CA, USA). Total protein (10 μg) was electrophoretically separated on sodium dodecyl sulfate-polyacrylamide gels (10%) and transferred to a polyvinylidene difluoride membranes (Amersham Hybond-P; GE Healthcare Life Sciences, Buckinghamshire, UK). The membranes were blocked with PVDF Blocking Reagent for CanGet Signal^®^ (Toyobo, Osaka, Japan) at room temperature for 1 h. The specific proteins on the membrane were immuno-reacted with rabbit antibodies (1/1000 dilution) against RAGE (Abcam), p38, phospho-p38, p44/42 ERK, phospho-ERK, p65-, and phospho-p65, and with β-actin rabbit antibody (1/10,000 dilution) at 4 °C overnight, and then reacted with horseradish peroxidase-conjugated goat anti-rabbit IgG (Cell Signaling) at room temperature for 1 h. The reacted membrane was developed using ECL Western Blotting Detection Reagents (GE Healthcare Life Sciences) and analyzed using Image Quant LAS 500 (GE Healthcare Japan, Life Sciences, Tokyo, Japan). The densities of bands of phosphorylated- and non- phosphorylated-p38, ERK, and p65 were determined using Image J software (NIH, Bethesda, MD, USA).

### 4.7. Enzyme-Linked Immunosorbent Assay

Sub-confluent HGFs were pre-treated with 6-shogaol (5 or 10 μM) for 1 h and cultured with AGEs (500 μg/mL) or BSA for 48 h. NAC (1 mM) was also added to HGF cultures with AGEs. The conditioned medium was collected and mixed with protease inhibitor cocktail, and cell lysates were extracted in lysis buffer including 10 mmol/l Tris-HCl, pH 7.4, 50 mmol/l NaCl, 5 mmol/l EDTA, 1 mmol/l sodium orthovanadate, 1% NP-40 and protease inhibitor cocktail. IL-6 in the conditioned medium was measured using a Human IL-6 ELISA kit (R&D Systems, Minneapolis, MN, USA) in accordance with the manufacturer’s instructions. ICAM-1 in the cell lysate was determined using a ICAM1 Human ELISA kit (Invitrogen, Carlsbad, CA, USA), and the amount of ICAM-1was normalized to total cell protein amount.

### 4.8. Statistical Analysis

All statistical analyses were performed with SPSS Statistics version 20 (IBM, Chicago, IL, USA). The significant differences among multiple groups were performed using one-way analysis of variance followed by Tukey’s HSD. *p* < 0.05 was considered significant.

## Figures and Tables

**Figure 1 molecules-24-03705-f001:**
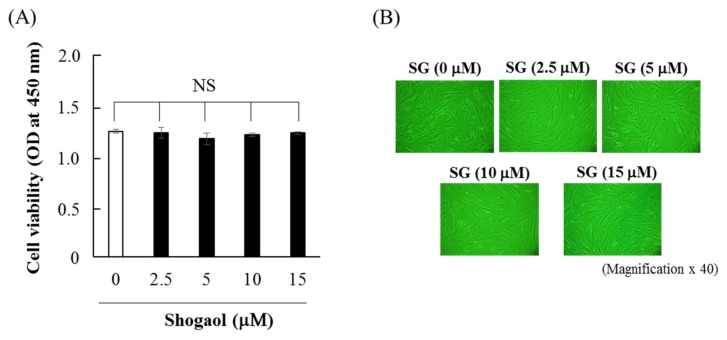
Effects of 6-shogaol on cell viability and the morphology of HGFs. HGFs were seeded at 4800 cells/cm^2^, cultured for 5 days, and then treated with 6-shogaol (2.5–15 μM) for 48 h. (**A**) Cell viability was assessed using Cell Counting Kit-8^®^. Data are expressed as the mean ± SD of 4 independent experiments. NS indicates no significant difference between the indicated groups. (**B**) Cultured HGFs were observed using phase-contrast microscopy after culture with 2.5–15 μM 6-shogaol for 48 h. (Magnification × 40).

**Figure 2 molecules-24-03705-f002:**
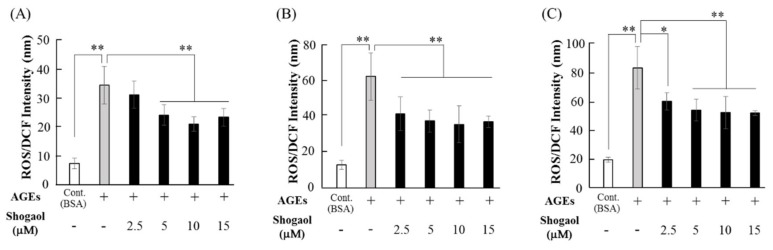
Effects of 6-shogaol on AGEs-induced ROS activity. Sub-confluent HGFs were pretreated with 6-shogaol (2.5–15 μM) *for 1 h and* cultured with AGEs (500 μg/mL) or BSA (500 μg/mL) for 12 h (**A**), 24 h (**B**), or 48 h (**C**). ROS activity was assessed by measuring a fluorescence intensity of 2′,7′-dichlorofluorescein (DCF) in HGFs treated with AGEs, BSA and 6-shogaol using an ROS activity assay kit as described in the Materials and Methods section. Data are expressed as the mean ± SD of 5 (**A**,**B**) and 4 (**C**) independent experiments (* *p* < 0.05, ** *p* < 0.01).

**Figure 3 molecules-24-03705-f003:**
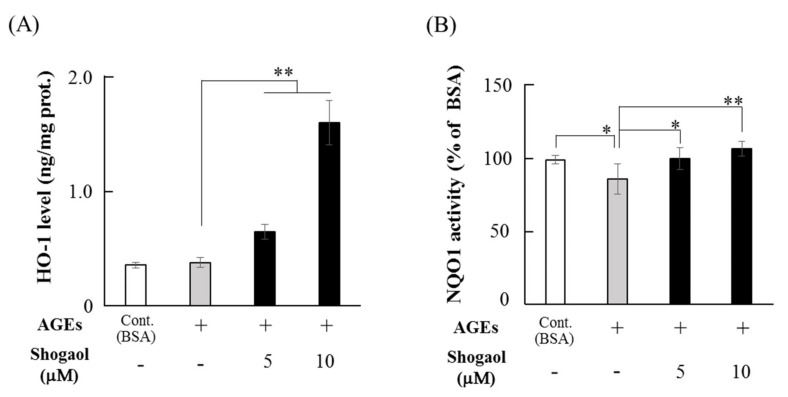
Effects of AGEs and 6-shogaol on antioxidant enzymes production. HGFs were cultured with AGEs (500 μg/mL) or BSA (500 μg/mL) for 12 h after pre-treatment with 6-shogaol (5 or 10 μM) for 1 h. (**A**) HO-1 levels in the cell lysate of treated HGFs were assessed using ELISA. (**B**) NQO1 levels in the cell lysate were assessed using an NQO1 activity assay kit, as described in the Materials and Methods section. Data are expressed as the mean ± SD of 6 (**A**) and 5 (**B**) independent experiments (* *p* < 0.05, ** *p* < 0.01).

**Figure 4 molecules-24-03705-f004:**
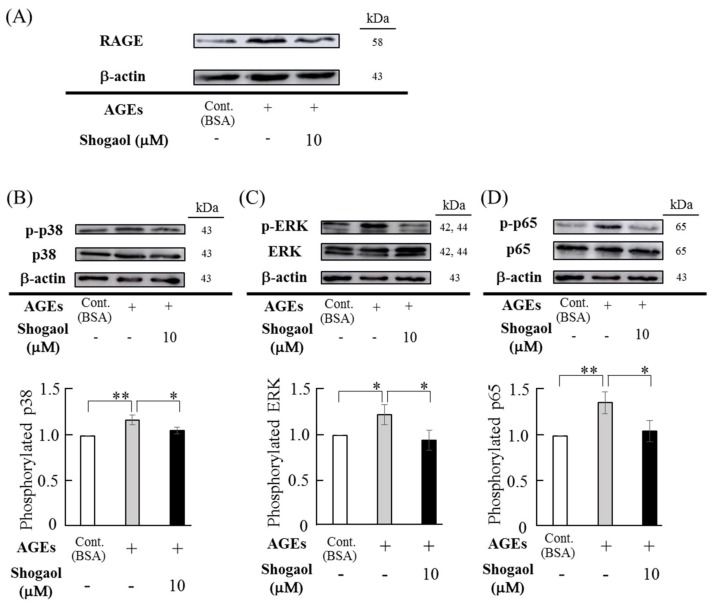
Effects of 6-shogaol on AGEs-induced RAGE expression and the activation of MAPK and NF-κB signaling pathways. (**A**) Sub-confluent HGFs were cultured with AGEs (500 μg/mL) or BSA (500 μg/mL) for 24 h after pre-treatment with 6-shogaol (10 μM) for 1 h. Cell lysates were used for western blotting for RAGE as described in the Materials and Methods section. (**B**–**D**) Sub-confluent HGFs were pre-treated with 6-shogaol (10 μM) and then stimulated with AGEs (500 μg/mL) or BSA (500 μg/mL) for 30 min. Cell lysates were analyzed by western blotting using antibodies against MAPKs (p38, phospho-p38, ERK and phospho-ERK) (**B**,**C**) and NF-κB (p65 and phospho-p65) (**D**) as described in the Materials and Methods section. Data are expressed as the mean ± SD of 3 independent experiments. The density of signal bands was determined by densitometric analysis and normalized to that of the BSA group. (* *p* < 0.05, ** *p* < 0.01).

**Figure 5 molecules-24-03705-f005:**
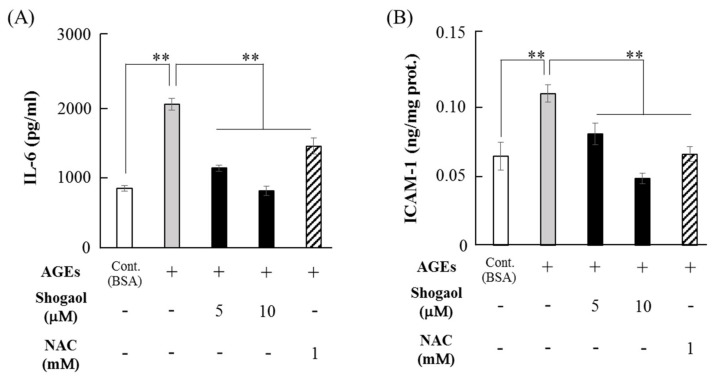
Effects of 6-shogaol on AGEs-induced production of IL-6 and ICAM-1. Sub-confluent HGFs were pre-treated with 6-shogaol (5 or 10 μM) for 1 h and cultured with AGEs (500 μg/mL) or BSA (500 μg/mL) for 48 h. NAC (1 mM) was also added into the HGF culture with AGEs. The amounts of IL-6 (**A**) in the supernatant and ICAM-1 (**B**) in the cell lysate were determined using their respective ELISA kits as described in the Materials and Methods section. Data are expressed as the mean ± SD of 6 independent experiments (** *p* < 0.01).

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
