# Peer review of "6-Shogaol Inhibits Advanced Glycation End-Products-Induced IL-6 and ICAM-1 Expression by Regulating Oxidative Responses in Human Gingival Fibroblasts"

_molecules, 2019, doi:10.3390/molecules24203705_

Round 1

Reviewer 1 Report

The authors have investigated the effects of 6-Shogaol on oxidative stress and AGEs-induced IL-6 and ICAM-1 expression in HGFs. The authors have reported that 6-Shogaol significantly inhibited AGEs-induced ROS activity, and increased antioxidant enzymes (HO-1, NQO1). In addition, the authors also reported 6-Shogaol decreased AGEs-induced ICAM-1 expression and phosphorylation of p38 and ERK MAPKs and p65. The data is very interesting for understanding of ginger extracts in the prevention and treatment of DM-associated periodontitis. However, the manuscript suffers from several weaknesses:

Why did the authors treat the HGFs with 0-15 µM of 6-Shogaol? The AGEs-induced ROS production still have higher ROS level after 6-Shogaol treatment in Fig 2. The author should explain this. What is the changes of ROS activity in HGFs cultured with 500 µg/ml of AGEs for 12h after pre-treated with 5 or 10 µM of 6-Shogaol? The authors should explain why only 10 µM of 6-Shogaol was used to investigate AGEs-induced RAGE expression and their signaling pathways. What is the loading control of western blotting in Fig 4B-D? Cell treatments were confused in Fig 1-5. Some letter is missing, such as “µ”.

Author Response

Reviewer 1

Q1: Why did the authors treat the HGFs with 0-15 μM of 6-Shogaol?

A1: We have referred shogaol concentrations which were used in others’ study including [36, 37, 46, 50]. And, we have used 2.5–15 μM of 6-shogaol in experiments of cell viability and ROS determination.

Q2: The AGEs-induced ROS production still have higher ROS level after 6-Shogaol treatment in Fig2. The authors should explain this.

A2: In the present study, 6-shogaol significantly decreased AGEs-induced ROS production in HGFs, however, its inhibition was not complete. Park et al. [46] also showed that 6-shogaol partially inhibited TNF-α+INF-γ-induced ROS production in human keratinocytes. We did not know the exact reason of partial inhibitory effect of 6-shogaol on ROS production. Since the regulation of ROS production by AGEs and 6-shogaol are associated with multiple factors including MAPK (p38, ERK, JNK), NF-κB and PI3K [12, 51-53], 6-shogaol may partially inhibit AGEs-induced ROS production. We have added this explanation in Discussion. (219-224)

Q3: What is the changes of ROS activity in HGFs cultured with 500 μg/ml of AGEs for 12h after pre-treated with 5 or 10 mM of 6-shogaol?

A3: 2.5 μM of 6-shogaol did not show a significant decrease of AGEs-induced ROS activity for 12 h treatment. In case of 24 and 48 h culture, 2.5 μM of 6-shogaol also significantly inhibited AGEs-induced ROS activity. This result showed that 6-shogaol at low concentration did not be effective for short, 12 h. We have added this explain in Results. (114-116)

Q4: The authors should explain why only 10 μM of 6-Shogaol was used to investigate AGEs-induced RAGE expression and their signaling pathways.

A4: We have used 10 μM of 6-shogaol in experiments of signaling pathways because 10 μM of 6-shogaol significantly inhibited AGE-induced ROS activity and increased HO-1 and NQO1 levels.  We confirmed that 10 and 15 μM of 6-shogaol inhibited AGEs-induced RAGE expression using a western blotting.

Furthermore, we have referred shogaol concentration which were used in others’ study including [36, 37, 46, 50]. They used 1-10 μM of 6-shogaol in their experiments using several cells.  

Q5: What is the loading control of western blotting in Fig 4B-D?

A5: We have added data of β-actin as a loading control in Fig 4B-4D.

Q6: Cell treatments were confused in Fig. 1-5.

A6: In experiments of Fig.2-5, we basically pretreated HGFs with 6-shogaol for 1h and then cultured with AGEs or BSA for 30 min (Fig.4B-4D), 12 h (Fig.2A, Fig.3), 24 h (Fig.2B, Fig.4A) and 48 h (Fig.2C, Fig.5). Although the time treated with factors were different due to examining factors including ROS, HO-1, NQO1, MAPK&NF-κB, IL-6 and ICAM-1, procedure of cell culture was basically similar. We have in part rewritten figure legend of Fig.2 to well understand this procedure. (120-123)

In experiment of Fig.1, HGFs were cultured with 6-shogaol for 48 h to investigate the effect of 6-shogaol on cell viability.

Q7: Some letter is missing, such as “μ”.

A7: We have correctly inputted “symbol” font in our manuscript.

Reviewer 2 Report

Review of manuscript entitled „6-Shogaol inhibits advanced glycation end-products - induced IL-6 and ICAM-1 expression by regulating  anti-oxidative activity in human gingival fibroblasts”

Tittle should be modified because is not is not adequate to given aim of these study or the aim of study should be more fit to the title.

Abstract

Abstract should be rewrote, because the aim of conducted studies is not well visible and it is done too much  general, well known information.

Introduction

Generally it would be advisable to edit the introduction, because in some places it is only a patch of separate information.

Line 34 – “Diabetes mellitus (DM) is a major risk factor of periodontal diseases.” – appropriate reference should be added.

Reference number [1] is too old and should be replaced by more adequate reference.

The most actual references should be given and construction of introduction and discussion section should be rethinking due to already known information, for example:

 “Advanced glycation end-products increase IL-6 and ICAM-1 expression via RAGE, MAPK and NF-κB pathways in human gingival fibroblasts. Nonaka K, Kajiura Y, Bando M, Sakamoto E, Inagaki Y, Lew JH, Naruishi K, Ikuta T, Yoshida K, Kobayashi T, Yoshie H, Nagata T, Kido J. J Periodontal Res. 2018 Jun;53(3):334-344. doi: 10.1111/jre.12518. Epub 2017 Nov 30.”

“EB 2017 Article: Changes in advanced glycation end products, beta-defensin-3, and interleukin-17 during diabetic periodontitis development in rhesus monkeys. Jiang H, Li Y, Ye C, Wu W, Liao G, Lu Y, Huang P. Exp Biol Med (Maywood). 2018 May;243(8):684-694. doi: 10.1177/1535370218766512. Epub 2018 Mar 27.”

and others

lines 46-47 – the information about “rat osteoblastic cells” is not necessary in this place.

Lines 47-49 -  What is the purpose of providing information on the results of previous authors' own research in this place??

Line 50 – what means “…several inflammatory diseases”? Additionally some information  are repeated in the lower lines.

Lines 56-57 – AGEs induced oxidative stress and increased ROS levels not only “in human umbilical vein endothelial cells 57 [19] and thyroid follicular epithelial cells [20]” as authors mention.

Line 66 – authors should be decided if “ROS is implicated in inflammation and the destruction of periodontal tissues with periodontitis [12].” Is only suggested or it is already known information, as was mentioned earlier.

Information given between lines 68-80 should be rather more useful in section discussion and in this place seems to be not clear connected with the title and aim of this study.

Lines 81-82 – reference [32} seems to be not adequate to given information, additionally what means “herbal medicine” – it concern any national or global medicine?

Line 97 – the information that “the effect of 6-shogaol on the pathogenesis of periodontitis with DM has not yet been elucidated” is rather exaggerated, because author did not examined pathogenesis process. This statement is not in accordance with title and the aim of this study.

The aim should be rewritten.

Results

The language and topographic mistakes are observed.

Authors should explain why they made different number of individual experiments.

2.1.

Why such concentration of 6-shogaol were used to experiments?

Figure 1B shows results quite weakly.

The using of BSA should be explain.

2.2.

The unit of used AGEs are somewhat mysterious. It should be compared to other literature date.

The symbols used on figures should be explain in the description of these figures.

2.3.

Authors should be explain why  “(bovine serum albumin [BSA])” was used as a control level and why 6-shogaol was used only in two concentrations (5 and 10 mM) as well as why such conditions of incubation were applied.

2.4.

The used concentration of 6-shogaol should be explain.

Lines 161-162 – authors should explain why they used AGE and BSA in such combination and during 30 minutes of incubation.

Line 179 – why authors added NAC (1 mM) into the HGF culture with AGEs? It should be explain.

Discussion

Line 184 – this first sentence should be rewrote.

Line 192 – it is repetition of information about trapping of MGO.

Lines 209 and 227  – an appropriate references should be added.

Generally discussion should be somewhat rewrote, especially the latest one sentence, because it is too general.

Materials and Methods

4.1.

Line 267 – how many times the PBS was changed?

Line 258 – it was really AGEs  activity measured?

Summarizing – after thorough re-editing, this manuscript will probably be suitable for publication.

Author Response

Reviewer 2

Title and Aim

Q1: Tittle should be modified because is not adequate to given aim of these study or the aim of study should be more fit to the title.

A1: We have partly modified a title and aim according to the suggestion of Reviewer 2. Title: (4) and Aim: (95-96).

Abstract

Q2: Abstract should be rewritten, because the aim of conducted studies is not well visible and it is done too much general, well known information.

A2: We have written Abstract to present their background, and to make clear aim and finding of our research and to suggest a usefulness of shogaol. (15-29)

Introduction

Generally it would be advisable to edit the introduction, because in some places it is only a patch of separate information.

Q3: Line 34- “Diabetes mellitus (DM) is a major risk factor of periodontal diseases.” – appropriate reference should be added.

A4: We have added an appropriate reference on line 34 as [1]. (354-355)

Q4: Reference number [1] is too old and should be replaced by more adequate reference.

A4: We have replaced a paper of AGEs with current paper on line 36 as new [2]

(356-357)

Q5: The most actual references should be given and construction of introduction and discussion section should be rethinking due to already known information, for example.

Lines 47-49 – What is the purpose of providing information on the results of previous authors’ own research in this place?

A5: We first performed a study to investigate the effect of AGEs on IL-6 and ICAM-1 expression in HGFs (Nonaka et al. J Periodontal Res 2018)[12] and had an idea of the present study from results of the first study. We think that the first paper is very important and should introduce the background related to the present study in Introduction. (46-48)

In contrast, our study using rat osteoblastic cells may not be suitable as a reference in Introduction. According to suggestion by the Reviewer 2, we have delated a previous paper and added Jiang’s paper. (42-43 and 368-370)

Q6: Line 50 – What means ”---several inflammatory diseases”? Additionally some information are repeated in the lower lines.

A6: We have moved a position of “[12-14]” after “periodontal diseases”, and written the detailed examples of “several inflammatory diseases” on the lower lines. (50-55)

Q7: Lines 56-57 –AGEs induced oxidative stress and increased ROS levels not only “in human umbilical vein endothelial cells 57 [19] and thyroid follicular epithelial cell [20]” as authors mention.

A7: The effects of AGEs on oxidative stress are well known, and we have shown typical two examples ([19] and [20] from some effects of AGEs on oxidative stress and ROS. To present more information about the effect of AGEs, we have added new reference. ([21] and 405-407)

Q8: Line 66 –authors should be decided if ROS is implicated in inflammation and the destruction of periodontal tissues with periodontitis [12].” Is only suggested or it is already known information, as was mentioned earlier.

A8: We wanted that the contents and references on 63-66 lines suggested “ROS is implicated in inflammation and the destruction of periodontal tissues with periodontitis”, and this content was described in [13]. However, because this relationship may be difficult to understand it, and we have rewritten “suggesting that” to “and” on lines 66-67.

Q9: Information given between lines 68-80 should be rather more useful in section discussion and in this place seems to not clear connected with the title and aim of this study.

A9: We think that information of antioxidative enzymes, and the relationship between those and periodontitis are necessary in Introduction. However, as the reviewer commented, these contents may be too much. Therefore, we have decreased the contents that is related to the present study. (68-76)

Q10: Lines 81-82 –Reference [32] seem to be not adequate to given information, additionally what means “herbal medicine” – it concern any national or global medicine?

A10: A review of ginger components, reference [32], introduces that ginger components function as medicine. However, we have deleted “medicine” and rewritten to “herbs” because our data did not directly show medical function. (78)

Q11: Line 97 – the information that “the effect of 6-shogaol on the pathogenesis of periodontitis with DM has not yet been elucidated” is rather exaggerated, because author did not examined pathogenesis process. This statement is not in accordance with title and the aim of this study.

A11: According to a comment by the reviewer, we have deleted “pathogenesis” and rewritten this sentence. (93-94)

Q12: The aim should be rewritten.

A12: As described in Q1, we have rewritten an aim that is closely associated with our study. (95-96)

Results

Q13: The language and topographic mistakes are observed.

A13: We have correctly converted symbol font in our manuscript.

Q14: Authors should explain why they made different number of individual experiments.

A14: The different number of experiments attribute to a proliferative character of the used cells. A proliferative character of human gingival fibroblast cell line, CRL-2014, was nervous. We have purchased this cell line from a cell bank at several times and used HGFs for each experiments. Although the number of experiments was different (from three to six), we think that these number of experiments is enough to evaluate results of experiments.

Q15: Figure 1: Why such concentration of 6-shogaol were used to experiments?

A15: We have referred shogaol concentrations which were used in others’ studies including [36, 37, 46, 50]. And, we have used 2.5–15 μM of 6-shogaol in experiments of cell viability and ROS determination.

Q16: Figure 1B shows results quite weakly.

A16: The picture of HGFs may be slightly difficult to evaluate the change of cellular morphology. However, we think that two examinations of cell viability and cellular morphology are very important to estimate cytotoxicity of 6-shogaol.

Q17: Figure 1: The using of BSA should be explain.

A17: We did not use BSA in an experiment of Figure 1 because we performed this experiment to investigate cytotoxicity of 6-shogaol.

Q18:Figure 2: The unit of used AGEs are somewhat mysterious. It should be compared to other literature date.

A18: We have chosen “500 μg/ml” as the optimum concentration of AGEs from a result in our previous study using the same HGFs (CRL-2014)(Ref. 12). We performed an experiment of AGEs dose-dependency (50-1000 μg/ml) and found that 500 μg/ml of AGEs significantly increased IL-6 expression in HGFs. We think that an optimum concentration of AGEs is dependent on cell species. Zhou et al. (Ref. 20) used 200 μg/ml of AGEs in human endothelial cells, and Hiroshima et al. used 500 μg/ml of AGEs in human gingival epithelial cells (J Cell Biochem 2018, 119:1591-1603).

Q20: Figure 2: The symbols used on figures should be explain in the description of these figures.

A20: “DCF” in graphs shows “2’,7’-dichlorofluorescein”. We determined ROS activity to measure fluorescence intensity of 2’,7’-dichlorofluorescein, and have added this explain in Materials and Methods, and figure legend of figure 2. (293, 295-296, 122-123)

Q21: Figure 3: Authors should be explain why “(bovine serum albumin [BSA])” was used as a control level and why 6-shogaol was used only in two concentrations (5 and 10 μM) as well as why such conditions of incubation were applied.

A21: “BSA” group shows that cells were treated with only BSA (without AGEs). We explained a role of BSA as control for AGEs in Materials and Methods because AGEs was prepared from BSA glycated with DL-glyceraldehyde (255-261). BSA is used as control many studies using AGEs, and has been used in the present experiments of Fig. 2, 3, 4 and 5, and described that in each figure legends. To make clear “BSA” as control, we have changed “BSA” to “Cont./BSA” in Figure 2, 3, 4 and 5.

We used 5 and 10 μM of 6-shogaol in two experiments of HO-1 and NQO1 because 6-shogaol at these concentrations significantly inhibited AGE-induced ROS activity (Figure 2). Other researchers showed that 5 and 10 μM of 6-shogaol up-regulated HO-1 level in human hepatoma cells and keratinocytes (Refs. 37 and 46).

Regarding the condition of incubation, we performed a similar procedure in Figure 2, 3, 4A and 5, and it consisted of the pretreatment with 6-shogaol for 1 h and treatment with AGEs or BSA for 12-48 h. We investigated the change of HO-1 level in HGFs cultured with factors for 12, 24 and 48 h. The similar change of HO-1 level was observed at 12 and 24 h as showing the following graphs.

Q23: Figure 4: The used concentration of 6-shogaol should be explain.

A23: We used 10 μM of 6-shogaol in experiments of Figure 4 because 6-shogaol at this concentration showed the effective functions in experiments of Figure 2 and Figure 3.

Q24: Figure 4: Lines 161-162 –authors should explain why they used AGE and BSA in such combination and during 30 minutes of incubation.

A24: We performed this experiment according to the procedure that was similar to the experiments of Figure 2 and 3. Therefore, HGFs were pretreated with 6-shogaol and then cultured with AGEs or BSA for 30 min.

We incubated HGFs with AGEs or BSA for 30 min according to the results of our previous study (Ref. 12).

Q25: Figure 5: Line 179 –why authors added NAC (1 mM) into the HGF culture with AGEs? It should be explain.

A25: We used NAC, an inhibitor of ROS, as a positive control for function of shogaol. We already wrote this explain in Results (169) and Discussion (237-239).

Discussion

Q26: Line 184 – this first sentence should be rewrote.

A26: We have rewritten this sentence (181-182).

Q27: Line 192 – it is repetition of information about trapping of MGO.

A27: The sentences on lines 182-187 introduces that 6-shogaol has two functions (inhibiting AGE-induced ROS, IL-6 and ICAM-1 expressions, and trapping MGO), and sentence on line 188-189 summarizes “6-shogaol has two functions”. We have partially rewritten this sentence to be well understood (188)

Q28: Lines 209 and 227 –an appropriate references should be added.

A28: We have inserted several references on lines 204 and 230, and we have described our speculation about function of 6-shogaol (205-206).

Q29: Generally discussion should be somewhat rewrote, especially the latest one sentence, because it is too general.

A29: We have summarized our study and described a medical possibility of ginger extracts (250-252).

Materials and Methods

Q30: Line 267?(256) –how many times the PBS was changed?

A30: We changed PBS at three times for 3 days to prepare AGEs. (259)

Q31: Line 258 –it was really AGEs activity measured?

A31: We have assessed AGEs activity by determining ratio of fluorescence intensity between AGEs and control non-glycated BSA according to with reference to the method of Huang et al. [62].

Regarding AGEs activity, we have set another our standards in study using HGFs. When we begin experiment using AGEs newly prepared, we check its dose dependence and then a responsibility of IL-6 expression to 500 μg/ml AGEs in HGFs. We confirmed more than 2-fold increase of IL-6 expression by AGEs in the previous [12] and present studies.

Round 2

Reviewer 1 Report

The revised manuscript has been improved from its original version, and the authors have addressed all my concerns.

Reviewer 2 Report

Authors improved their manuscript. In present form is suitable fro publication.